# Cytokines and Lymphoid Populations as Potential Biomarkers in Locally and Borderline Pancreatic Adenocarcinoma

**DOI:** 10.3390/cancers14235993

**Published:** 2022-12-05

**Authors:** Iranzu González-Borja, Antonio Viúdez, Emilia Alors-Pérez, Saioa Goñi, Irene Amat, Ismael Ghanem, Roberto Pazo-Cid, Jaime Feliu, Laura Alonso, Carlos López, Virginia Arrazubi, Javier Gallego, Jairo Pérez-Sanz, Irene Hernández-García, Ruth Vera, Justo P Castaño, Joaquín Fernández-Irigoyen

**Affiliations:** 1OncobionaTras Lab, Navarrabiomed, Navarra University Hospital, Universidad Pública de Navarra (UPNA), 31006 Pamplona, Spain; 2Medical Oncology Department, Navarra University Hospital, 31008 Pamplona, Spain; 3Maimonides Biomedical Research Institute of Córdoba, 14004 Córdoba, Spain; 4Department of Cell Biology, Physiology, and Immunology, University of Córdoba, 14071 Córdoba, Spain; 5Reina Sofía University Hospital, 14004 Córdoba, Spain; 6Centro de Investigación Biomédica en Red de Fisiopatología de la Obesidad y Nutrición, (CIBERobn), 14004 Córdoba, Spain; 7Pathology Department, Navarra University Hospital, 31008 Pamplona, Spain; 8Medical Oncology Department, La Paz University Hospital, 28046 Madrid, Spain; 9Medical Oncology Department, Miguel Servet University Hospital, 50009 Zaragoza, Spain; 10Medical Oncology Department, Marqués de Valdecilla University Hospital, 39008 Santander, Spain; 11Medical Oncology Department, Hospital General Universitario de Elche, 03203 Elche, Spain; 12Clinical Neuroproteomics Unit, Navarrabiomed, Proteored-ISCIII, Proteomics Unit, Navarrabiomed, Navarra University Hospital, Universidad Pública de Navarra (UPNA), Instituto de Investigación Sanitaria de Navarra (IdiSNA), 31008 Pamplona, Spain

**Keywords:** pancreatic ductal adenocarcinoma (PDAC), biomarkers, resectable disease, borderline disease, cytokines and growth factors, T lymphocytes, B lymphocytes, protein arrays, flow cytometry and immunohistochemistry

## Abstract

**Simple Summary:**

PDAC remains as one of the deadliest types of cancer due to its late diagnosis, its inherent aggressiveness, and the low efficacy of routinely used treatments (from surgery or radiotherapy to systemic treatments). The search of new potential biomarkers is paramount in this context, where CA19-9 is still the only recommended biomarker for the management of this disease. Thus, the main goal of the present study was to assess the potential value as predictive/prognostic biomarkers of several cytokines and growth factors in serum, as well as circulating immune populations in a cohort of 64 PDAC patients.

**Abstract:**

Despite its relative low incidence, PDAC is one of the most aggressive and lethal types of cancer, being currently the seventh leading cause of cancer death worldwide, with a 5-year survival rate of 10.8%. Taking into consideration the necessity to improve the prognosis of these patients, this research has been focused on the discovery of new biomarkers. For this purpose, patients with BL and resectable disease were recruited. Serum cytokines and growth factors were monitored at different time points using protein arrays. Immune cell populations were determined by flow cytometry in peripheral blood as well as by immunohistochemistry (IHC) in tumor tissues. Several cytokines were found to be differentially expressed between the study subgroups. In the BL disease setting, two different scores were proven to be independent prognostic factors for progression-free survival (PFS) (based on IL-10, MDC, MIF, and eotaxin-3) and OS (based on eotaxin-3, NT-3, FGF-9, and IP10). In the same context, CA19-9 was found to play a role as independent prognostic factor for OS. Eotaxin-3 and MDC cytokines for PFS, and eotaxin-3, NT-3, and CKβ8-1 for OS, were shown to be predictive biomarkers for *nab*-paclitaxel and gemcitabine regimen. Similarly, oncostatin, BDNF, and IP10 cytokines were proven to act as predictive biomarkers regarding PFS, for FOLFIRINOX regimen. In the resectable cohort, RANTES, TIMP-1, FGF-4, and IL-10 individually differentiated patients according to their cancer-associated survival. Regarding immune cell populations, baseline high levels of circulating B lymphocytes were related to a significantly longer OS, while these levels significantly decreased as progression occurred. Similarly, baseline high levels of helper lymphocytes (CD4+), low levels of cytotoxic lymphocytes (CD8+), and a high CD4/CD8 ratio, were related to a significantly longer PFS. Finally, high levels of CD4+ and CD8+ intratumoural infiltration was associated with significantly longer PFS. In conclusion, in this study we were able to identify several prognostic and predictive biomarker candidates in patients diagnosed of resectable or BL PDAC.

## 1. Introduction

Pancreatic ductal adenocarcinoma (PDAC) currently ranks as the seventh leading cause of cancer-related deaths worldwide [1,2], having very similar incidence and mortality rates [3]. This generally relates to the fact that at the time of diagnosis, around 80% of patients are at an advanced clinical-radiological stage. The absence of specific biomarkers apart from carbohydrate antigen 19-9 (CA19-9), which is still considered as the unique recommendation by the American and European guidelines for diagnosis and follow-up of potentially resectable cases, and the non-specific signs and symptoms during the initial stages of the disease, are the reason of the latter [4,5].

Historically, cytokines have been examined in different types of neoplasia with diagnostic and therapeutic purposes, as they are one of the major modulators of the immune system, due to their pleiotropic nature [6]. Since Cohen introduced the term “cytokine” for the first time in 1974, several studies have analyzed the role of these molecules as novel diagnostic, predictive, and/or prognostic biomarkers in PDAC [7,8,9,10,11,12,13].

Compilation of data obtained from cytokine arrays enables the characterization of PDAC disease, in order to design therapeutic protocols for PDAC treatment. In this regard, several clinical trials intend to design different strategies, following the inhibition of some of these cytokines that would promote PDAC onset or reduce life expectancy [14]. In this line, a recent study evaluated the therapeutic effect of blocking IL-20 by 7E (anti- IL-20 monoclonal antibody) in an KPC (LSL-Kras^G12D^; Trp53^flox/flox^; Pdx-1-Cre) PDAC mouse model and an KPC cell-injected orthotopic model, resulting in a prolonged survival of mice, partially explained by the attenuation of PD-L1 expression in both models. The use of 7E also reduced the polarization of macrophages to M2 phenotype, reducing their tumor infiltration [15].

From the therapeutic perspective IL-2, IL-15, IL-21, IL-12, GM-CSF, and IFN-α among others have been tested in several clinical trials [14]. Despite these strenuous efforts, only two of these cytokines have been approved by the FDA, IFN-α for the treatment of hairy cell leukemia since 1986 or as adjuvant treatment for melanoma, and IL-2 for the treatment of metastatic renal cancer and advanced melanoma since 1992 and 1998, respectively [14]. One of the major concerns regarding their use is related to their pleiotropic behavior, as well as their redundancy or biological promiscuity, which may partly explain their low efficacy profile and the considerable rate of side effects. This complex scenario has undoubtedly facilitated the replacement of this type of immunomodulators by more specific targeted therapies and other types of immunotherapies with a greater efficacy and better toxicity profile. In this regard, the COMBAT trial tested the efficacy of BL-8040, a specific CXCR4 antagonist, in combination with pembrolizumab and standard chemotherapy in PDAC [16]. Surprisingly for this tumor entity, the triple therapy showed an objective response rate (ORR) of more than 30% and a disease control rate (DCR) on 3 out of 4 subjects, with a median duration of response of 7.8 months, which indicated improved results when indirectly compared with the only currently approved second-line regimen for metastatic PDAC [16]. Similarly, the first results of a phase I/II clinical trial with olaptesed (NOX-A12), a specific CXCL12 inhibitor, were recently presented in heavily pre-treated PDAC and metastatic colorectal adenocarcinoma patients, concluding that 25% of patients achieved stable disease and long-term disease control by an increase of effector immune cell infiltration of the tumor (NCT03168139) [17].

On the other hand, there is a growing interest on the analysis of circulating and infiltrating immune cell populations in different neoplasia such as hepatocellular carcinoma, colorectal cancer, follicular lymphoma, acute myeloid leukemia, and PDAC [18,19,20,21]. In PDAC setting for instance, levels of CD3^+^, the helper (CD4^+^)/cytotoxic (CD8^+^) ratio, and CD8^+^CD28^+^ populations were able to discriminate patients with PDAC from those with benign disease or healthy donors. In resectable PDAC, CD8^+^CD28^+^ cell population has been recently shown as an independent prognostic factor for OS [22]. Importantly, a decreased ratio of circulating CD4/CD8 after two cycles of conventional chemotherapy may suggest an improved OS in different settings of PDAC [23]. As in other tumor entities, the histologic response to previous administered systemic therapy in PDAC, was clearly related to an increased number of infiltrating CD8^+^ compared with upfront surgery patients [24].

Regarding other immune populations, B-cell tumor infiltration has been associated with a better prognosis in several tumors (breast, colorectal, NSCLC, head and neck, ovarian, biliary tract, melanoma, and liver) and its predictive value has recently been observed in melanoma when combination of pembrolizumab and ipilimumab was used [25]. Regarding pancreatic neoplasms, there are scarce and contradictory studies characterizing the role of B lymphocytes in this setting. For instance, a study including 160 resectable PDAC patients, demonstrated that in those with lower B-cell levels had a statistically significant longer OS, with a clear role as an independent prognostic factor [22]. Discordantly, another study with 73 subjects with pancreatic neuroendocrine tumors showed that high levels of B cells were associated with longer PFS. These contradictory results might be partially explained by the different functions of B lymphocytes, since they can induce an anti-tumor response (generating tertiary lymphoid organs, production of antibodies and/or antigen presentation), or a pro-tumor response (promoting tumor growth, aberrant tumor angiogenesis, activation of myeloid derived suppressor cells-MDSCs- and preventing lymphocyte responses) [26].

Based on the previous rational and background, in this study we have focused on the search of predictive and/or prognostic biomarkers, characterizing the circulating and tissue infiltrating T and B lymphocyte populations, and serum cytokines and growth factors, in the setting of BL and resectable PDAC tumors in whom surgery was planned as the main approach.

## 2. Materials and Methods

In this multicenter and prospective study, subjects with BL and resectable PDAC were enrolled. The study was approved by the Ethics Committee of the Government of Navarre-Spain (Pyto 2017/69) and The Spanish Agency of Medicine and Medical Devices-AEMPS (AVB-NAB-2018-01). Eligible patients for the BL cohort were patients with 18 years old or older, diagnosed with BL resectable PDAC (II-III stages), based on NCCN criteria [27], with poor prognostic factors (suspected of micrometastatic disease, risk of positive margin during surgery and/or CA 19.9 ≥ 300 UI/mL without jaundice). In the second cohort, eligible subjects were those with suspected PDAC that were considered as initially resectable by consensus of a multidisciplinary committee.

### 2.1. Human Samples

Peripheral blood samples were obtained throughout patient follow-up; before starting any type of systemic treatment, after radiological reassessment (the radiological assessment was performed after neoadjuvant treatment, after 2–3 months of this neoadjuvant treatment), 48 h after surgical procedure, and a year after surgery or at radiological progression of the disease (whichever happened earlier). Blood was collected in EDTA containing tubes (Greiner Bio-One, Kremsmünster, Austria) for plasma and immune cell population isolation, and in clot accelerator tubes Z Serum Sep Clot Activator (Greiner Bio-One, Kremsmünster, Austria) for serum isolation. All blood samples were processed freshly. Tumor samples were obtained only from those patients that could be resected surgically.

### 2.2. Human Cytokine Antibody Array

Cytokines and/or growth factors were analyzed by a protein array (Human Cytokine Antibody array −80 Targets-, ab133998, Abcam, UK) following manufacturer’s recommendations.

Briefly, the protein content of serum samples was quantified using Bradford reagent. Membranes were blocked with blocking buffer at room temperature (RT) for 30 min and incubated overnight at 4 °C with 100 µg of serum protein diluted in 1 mL blocking buffer. The next steps include two overnight incubations, at 4 °C, with biotinylated anti-cytokine antibody mix following HRP-conjugated streptavidin, with serial washing of the membranes between each incubation step. Finally, membranes were incubated with detection buffer for 2 min and revealed by Chemidoc^TM^ MP Imaging System (Bio-Rad, Hercules, CA, USA). Signals were quantified using Image Lab^TM^ software (Bio-Rad) and normalized using the six positive controls present on each membrane, using the mean of a control group to normalize the array data. To normalize the values, the following calculus was performed:X(Ny) = X(y) ∗ P1/P(y)
where X (Ny) = normalized signal intensity for spot “x” on array “y”; X(y) = mean signal density for spot “x” on array for sample “y”; P1 = mean signal density of 6 positive control spots of healthy donors; P(y) = mean signal density of positive control spots on array “y”. After this normalization, the relative expression levels of each cytokine in our samples of interest were compared. For statistical analysis Perseus software (Version 1.6.5) [28] was used, where the previously obtained normalized data “log2” was applied. Then the data were normalized with adjustment, and the correspondent statistical test as two-way Student´s *t*-test between groups was performed. The *p*-values and fold changes obtained were then represented with the Scatter Plot tool.

### 2.3. Flow Cytometry

The immune populations of interest were isolated from peripheral blood using Ficoll density gradient (17-1440-02, GE Healthcare, Marlborough, MA, USA) and purified for labelling with fluorophore-conjugated antibodies, are detailed in Appendix A. T cells were classified according to CD27/CD28 expression markers into poorly differentiated (CD27^+^ CD28^+^, TPD), intermediately differentiated (CD27negative CD28^+^, TID), and highly differentiated (CD27negative CD28low/negative, THD) subsets. B cells were characterized as CD11b^−^ CD19^+^.

The cells were examined in a FACSCanto^TM^ II flow cytometer (BD-Becton Dickinson, Franklin Lakes, NJ, USA) equipped with FACSDiva v.6.0 software (BD-Becton Dickinson, NJ, USA). The analysis was performed using FlowJo Software (FlowJo^TM^ Version 10. Ashland, OR: Becton, Dickinson and Company; 2019).

### 2.4. Immunohistochemistry

CD4 and CD8 IHC was performed at the Pathology Department of the HUN, in the automatic Leica Bond Max device (Leica Biosystems, Wetzlar, Germany), using Refine 15 protocol and FFPE slides from human specimens. The IHC stains were analyzed by expert pathologists, following the criteria used in the clinical setting. Briefly, all IHC were observed under the microscopy and three intensity groups were considered, according to the area of CD4 or CD8 infiltration. The categories were the following: low, intermediate, and high infiltration. Antibodies are detailed in Appendix A.

### 2.5. Data Collection and Statistical Analysis

Assuming a 95% confidence level, a drop-out hazard of 0.1 per person-year, and a relapse rate of 0.3 per person-year for the neoadjuvant group, the sample size needed to detect as significant, with 80% power, a rate ratio for the neoadjuvant group with favorable outcome with respect to that with unfavorable outcome of 2.25, the sample size will be 44 neoadjuvant-treated patients. Since no baseline data are available for the case of adjuvant treatment and considering that the disease-free survival of these patients could be similar to that PFS for neoadjuvant subjects, around 15 patients will likewise be recruited to perform the study in this cohort. Calculations were performed with the gsDesign library of the R program, 2.3.1.

Regarding the clinical and demographic information, frequencies were expressed in percentage and chi-square test and/or Fisher exact test were used to compare both cohorts of patients. Event time distributions for disease-free survival (DFS), progression-free survival (PFS), overall survival (OS), and cancer-associated death were estimated with Kaplan–Meier method and compared using the Log-rank statistic test or the Cox proportional-hazards regression model. Variables shown by univariate analysis to be significantly associated with DFS, PFS, or OS were entered into a Cox proportional hazards regression model for multivariate analysis. Those cytokines/and immune subsets in peripheral blood that showed statistical significance in Log-rank analysis were dichotomized and clustered in scores filtered by HR (>0.5) to avoid co-linearity effect during the multivariate analyses (Appendix A).

Statistical analysis was performed with non-parametric statistical tests; Kruskal–Wallis for independent samples with non-parametric distribution with Dunn’s multiple comparison test; Wilcoxon matched-pairs signed-rank test was used as a two-paired samples; and Mann–Whitney *U*-test for two non-paired samples. For experiments following normal distribution, ANOVA and Two-way Student’s t-test were performed.

Statistical tests were performed with IBM SPSS Statistics for Windows, Version 21 (Armonk, NY: IBM Corp). Other graphical representations were done using GraphPad Prism version 8.0.2 (GraphPad Software, San Diego, CA, USA, www.graphpad.com (accessed on 23 May 2022)), with mean values and standard deviation. The mean of each variable including cytokines and immune cell populations was used as cut-off value for survival analyses, “low” and “high” were assigned to levels below and above this cut-off value, respectively.

## 3. Results

### 3.1. Clinical and Demographic Data

Sixty-four patients diagnosed of PDAC disease were finally enrolled in this study, forty-seven of them considered as BL disease and seventeen with resectable disease, with a median follow-up of 12.48 months (4.67–49.25 months). Clinical and demographic data of PDAC patients are summarized in Table 1.

In the BL group, those patients who underwent surgery had significantly longer PFS (13.40 vs. 4.42 months, *p* = 0.0001). Similarly, those with pre-surgery serum CA19-9 levels below 100 U/mL showed significantly longer PFS (12.94 vs. 7.12 months, *p* = 0.001). Regarding the type of chemotherapy received, no statistical differences were observed between FOLFIRINOX and *Nab*-paclitaxel- gemcitabine regimens (Appendix A). In the resectable cohort, those patients with pT1-T2 exhibited longer DFS in comparison with pT3-T4 patients (14.58 vs. 8.44 months, *p* = 0.025). A non-significant trend for longer DFS was also observed in those cases with an absence of positive lymph nodes (21.71 vs. 8.44 months, *p* = 0.067).

### 3.2. Serum Cytokine Levels Are Correlated with Clinical Outcome in the BL Cohort

Baseline cytokine levels in the BL cohort revealed statistically significant differences in terms of survival analyses for eotaxin-3, FGF-9, IP10, NT-3, IL10, MDC, and MIF independently *(data not shown).* Subsequently, two scores based on these cytokines were developed using IL-10, MDC, MIF, and eotaxin-3 for PFS analysis (Figure 1A) and eotaxin-3, NT-3, FGF-9, and IP-10 for OS analysis (Figure 1B).

Additionally, systemic cytokine profiles were also compared between a selection of subjects considered as responders (R, *n* = 11) or non-responders (NR, *n* = 9), according to the radiological assessment. The radiological assessment was performed after neoadjuvant treatment, after 2–3 months of this neoadjuvant treatment. The comparison proved that IL-10 and interleukin 1-beta (IL-1β) were significantly under-expressed in R compared to NR (*p* = 0.017 and *p* = 0.034, respectively) (this analysis was only considering the basal timepoint) (Figure 2). The kinetics of IL-10 and IL-1β at baseline and radiological test were represented, and differences were observed between R and NR (*p* = 0.003 in both comparison) (Figure 2).

In addition, low levels of IL-1β were significantly correlated with higher pT stages (*p* = 0.041) and IV dissemination (*p* = 0.020), without significant impact in PFS and OS). Low levels of IL-10 were significantly correlated with clinical N0 (*p* = 0.029).

In the *nab*-paclitaxel- and gemcitabine-receiving sub cohort, high levels of eotaxin-3 and low levels of MDC were associated with longer PFS (HR = 0.31; 95% CI: 0.11–0.85; *p* = 0.017 and HR = 0.28; 95% CI: 0.11–0.73; *p* = 0.006, respectively) (Figure 3A–C). In the same line, high levels of eotaxin-3 and NT-3 levels, and low levels of CKβ8-1 were associated with longer OS (HR = 0.39; 95% CI: 0.16–0.95; *p* = 0.034, HR = 0.28; 95% CI: 0.12–0.90; *p* = 0.040, and HR = 0.29; 95% CI: 0.10–0.86; *p* = 0.025, respectively) (Figure 3D–E). Finally, low levels of angiogenin were significantly associated with longer cancer-associated death time (HR = 0.44; 95% CI: 0.17–1.10; *p* = 0.038) (Figure 3F).

In the FOLFIRINOX receiving sub cohort patients with low levels of oncostatin, BDNF and IP10 reached significantly longer PFS (HR = 0.11; 95% CI: 0.11–1.08; *p* = 0.037, HR = 0.13; 95% CI: 0.14–1.34; *p* = 0.046, and HR = 0.11; 95% CI: 0.11–1.08; *p* = 0.037, respectively). Cancer-associated death time was significantly longer in subjects with low levels of oncostatin, IP10 or NAP-2 (HR = 0.003; 95% CI: 0–925; *p* = 0.004, HR = 0.003; 95% CI: 0–925; *p* = 0.004, and HR = 0.014; 95% CI: 0–45; *p* = 0.04, respectively) (Figure 3G–L).

### 3.3. Serum Cytokine Levels Are Correlated with Clinical Outcome in the Resectable Cohort of Patients

Cancer-associated death was significantly longer in subjects with low levels of RANTES, TIMP-1, and FGF-4 (*p* = 0.002, *p* = 0.014 and *p* = 0.028, respectively). In contrast, patients with high levels of IL-10 achieved longer OS (27.86 months vs. 10.54 months, *p* = 0.028) (Figure 4).

### 3.4. Circulating Immune Population Frequencies and Cytokine Levels Are Correlated

Baseline circulating levels of B cells were found to be correlated with some serum cytokines, being positively correlated with I-309 and inversely correlated with FGF-6, NAP-2 and TGF-beta2 (Appendix A).

Similarly, circulating CD4/CD8 ratio was positively correlated with IGF-I, Flt-3 ligand, IP-10, and RANTES and inversely correlated with IL-10, HGF, Eotaxin-2, IGFBP-1, and IGFBP-3 (Appendix A).

### 3.5. Circulating B and T Cell Populations Are Associated with Clinical Outcome

High baseline circulating B cells frequency was related to a significantly longer OS in the BL cohort of patients (*p* = 0.033) (Figure 5A). Importantly, B-cell population was significantly decreased in comparison with baseline samples (*p* = 0.044) when progression disease was confirmed (Figure 5B).

Regarding T lymphocytes, different subsets were compared in healthy and BL cohorts. Statistically significant differences were observed in CD4^+^ T cells, CD8^+^ T cells, CD4 T_PD_ cells, and CD8 T_ID_ cells (Appendix A). High levels of circulating CD3^+^ T cells and CD4^+^ T cells were associated with a longer PFS (Figure 5C,D), while having low levels of CD8^+^ T cells was associated with longer PFS (Figure 5E). Consistently, a higher CD4/CD8 ratio was associated with longer PFS (14.06 months vs. 8.50 months, *p* = 0.004) (Figure 5F). Density plot graphs of B cell and CD4+/CD8+ T cells frequencies comparing patients with favorable and unfavorable outcome showed differences at first sight (Appendix A).

Strikingly, most of the patients with higher levels of CD4^+^ T_ID_ were established as cT1 and cT2 (*p* = 0.030). In line with this patients with higher CD8^+^ T_HD_ were associated with clinical N0 (*p* = 0.012). No statistical differences were appreciated in the survival analysis according to different T-cell *subsets*.

### 3.6. Tumour Infiltrating CD4 and CD8 T Lymphocytes Correlate with Clinical Outcome in BL Cohort

Higher levels of tumor infiltrating CD4^+^ T cells and CD8^+^ T cells were associated with a significantly longer PFS (*p* = 0.022 and *p* = 0.043, respectively) (Figure 6A,B). Moreover, higher infiltration of CD8^+^ T cells was significantly related to a longer OS and cancer-associated survival (*p* = 0.019 and *p* = 0.023, respectively) (Figure 6C,D). T-cell infiltration was most commonly observed in the area surrounding the tumor, with a higher proportion of CD8+ T cells, compared to CD4+ T cells (Appendix A). No other association with other clinical-pathological parameters was seen. No association was observed between the percentages of circulating immune populations and tumor infiltration in the eight matched cases analyzed (*p* = 1 for CD4^+^ T and *p* = 0.143 for CD8^+^T).

### 3.7. Multivariable Analysis

In the BL cohort, our newly generated score (based on IL-10, MDC, MIF, and eotaxin-3 cytokines) showed a significant role as independent prognostic factor for PFS (HR = 4.07; 95% CI: 1.48–11.5; *p* = 0.006). Comparably, CA19-9 levels at diagnosis (HR = 0.101; 95% CI: 0.01–0.94; *p* = 0.044) and the second generated score (based on eotaxin-3, NT-3, FGF-9 and IP-10) (HR = 4.80; 95% CI 1.03–22.40; *p* = 0.046) played similar roles as independent prognostic factors for OS.

## 4. Discussion

In 2020, PDAC accounted for around 2.6% of tumors diagnosed worldwide, being considered the seventh leading cause of cancer deaths (4.7%). This is mostly explained by an extremely comparable annual reported incidence and mortality rate (being in 2021 of 495,773 and 466,003 subjects, respectively) [3]. This dramatic situation highlights the urgent need to find solid diagnostic, prognostic, and predictive biomarkers that will help to improve these parameters. Nowadays, it is widely accepted that tumor niche-specific cytokines and immune populations are able to enter the bloodstream, representing a potential source of tumor-associated biological material that can be dynamically and easily analyzed.

The comparison of systemic cytokines between responders and non-responders disclosed two cytokines, IL-10 and IL-1β, that were significantly under-expressed in the first cluster, with opposite kinetic behavior between responders and non-responders. Both cytokines have pro- and anti-tumor properties, being IL-10 widely associated with worse prognosis, not only in solid tumors but also hematological malignancies [29,30,31]. In line with this, BL patients with lower IL-10 levels showed statistically significant longer PFS. Yet, we obtained contradictory results in the resectable cohort, regarding cancer-associated survival. Other biological functions of IL-10 include activation of T and NK cells, hence showing dual immunosuppressive and immunostimulatory activity [32]. Accordingly, such immunological ambivalence may be influenced by the surgical procedure of removing the primary tumor. The research over a larger cohort of patients without neoadjuvant treatment might support these findings. On the other hand, high expression of IL-1β was associated with poor OS in PDAC patients [33]. Beyond prognosis, IL-1β can also influence anti-cancer treatments. In fact, chemotherapy and radiation can induce the production of IL-1β by either cancer cells or tumor infiltrating cells [34]. Based on our results, we suggest that IL-10 and IL-1β could identify patients with an increased probability to respond and, therefore, to have a better prognosis, regardless the treatment received. This finding raises the question about the hypothetical use of IL-10 and IL-1β inhibitors in cancer treatment, with the purpose of enhancing the therapeutic effect of commonly used ordinary treatments.

Cytokines may have a potential role as survival predictors to the treatment received, as in was shown in our two cohorts of subjects treated with *nab*-paclitaxel and gemcitabine, and FOLFIRINOX. In this regard, and even though IP-10/CXCL10 has an influence on the trafficking of autoaggressive cells during development of several autoimmune diseases, mainly type 1 diabetes [35], to our knowledge this was the first time that IP-10/CXCL10’s predictive capacity regarding PFS and cancer-associated mortality was demonstrated in FOLFIRINOX-treated subcohort. Likewise, eotaxin-3 levels, without a clear impact in carcinogenesis beyond a singular previous reference regarding its predictive role in patients with melanoma treated with anti-PD1 drugs [36], was demonstrated to act as predictive factor for PFS and OS in *nab*-paclitaxel and gemcitabine treated subcohort. High plasma levels of angiogenin (EGFR ligand) have been associated with erlotinib (EGFR inhibitor) sensitivity [37], although clinically questionable benefit was seen when combined with gemcitabine. In contrast and based on our results, low levels of angiogenin were associated with increased cancer-associated survival in BL cohort treated with *nab*-paclitaxel and gemcitabine.

Currently, systemic characterization of different immune populations is being carried out in a plethora of different types of cancer [18,19,20,21], although inconsistencies are found in PDAC [22,23,38,39], especially for B lymphocytes [22,40]. In resectable PDAC, having circulating low B-cell levels was associated with longer OS, being a strong independent prognostic factor [22]. Contradictorily, higher B-cell levels in pancreatic neuroendocrine tumors were associated with longer PFS [40], in coincidence with our results in BL PDAC OS. The present study may be the first study that has been able to evaluate circulating B-cell levels in patients with BL PDAC, showing an evident difference concerning survival. Moreover, our dynamic determinations demonstrated that subjects under progression disease showed a decrease in B cells compared to baseline levels, which could be interpreted as a classic behavior of tumor immunosuppression. Supporting this discovery, a prospective study of 26 cancer patients, including cases with PDAC, showed a significant decrease of B cells in peripheral blood in patients with poor response to chemotherapy [41]. This immunosuppression phenomenon is characterized by a decrease in B cells, but not in other immune cells such as NK or T lymphocytes. Hence the individualized assessment of B cells could be, in itself, of great clinical value. Finally to complete our approach, it could be interesting to assess B-cell infiltration of tumor tissues by IHC, as although this subpopulation could represent around 25% of TILs in cancer; fewer studies have addressed this [26]. B-lymphocyte infiltration has been associated with a better prognosis in different tumors, including PDAC, with an intelligible predictive value in pembrolizumab and ipilimumab combination in melanoma [25]. Drugs such as ibrutinib have been tested in different animal models of PDAC, showing a decrease in tumor size and improved overall survival [42]. However, in the recent RESOLVE clinical trial where the combination of ibrutinib and *nab*-paclitaxel/gemcitabine in stage IV PDAC was tested, the data of PFS and OS were disappointing [43]. Despite these previous results, there is a field of development of new- targeted therapies as well as an urgent need for further studies testing the role of B cells and their subpopulations.

With respect to T lymphocytes, our baseline results indicate that patients with higher circulating total lymphocyte levels, as well as CD4+ lymphocytes had a significant longer PFS, opposite to what is seen for CD8+ lymphocytes. In line with this, subjects with a high CD4+/CD8+ ratio, showed a significant increase in terms of PFS. Taking into consideration that preceding similar results have been published in pNET [40] and colorectal cancer [19], but not in BL PDAC, our results suggested for the first time that a higher tumor infiltration of both CD4+ and CD8+ cell subtypes was associated with higher PFS, with a substantial increase in OS and cancer-associated survival in case of high CD8+ lymphocytes. Likewise, another recent study demonstrated that the response to neoadjuvant treatment in BL patients was characterized by an enhanced infiltration of CD8+ lymphocytes compared to patients resected from the beginning [24].

Finally, and only focusing on the BL cohort, we were able to develop a score based on circulating IL-10, MDC, MIF, and eotaxin-3 levels that showed a meaningful statistically significant prognostic role for PFS (HR = 4.07; 95% CI: 1.48–1.15; *p* = 0.006). Similarly, another score based on circulating eotaxin-3, NT-3, FGF-9, and IP-10 levels was established as an independent prognostic factor for OS (HR = 4.80; 95% CI 1.03–22.40; *p* = 0.046). In parallel, pre-surgery CA19-9 levels were also established as an independent prognostic factor for OS (HR = 0.101; 95% CI: 0.01–0.94; *p* = 0.044), in concordance with what previously was established in several studies [44,45,46]. Nevertheless, and to the best of our knowledge, there is no previous evidence that similar scores based on categorical cytokine variables determined by arrays could have a prognostic influence, in terms of PFS and/or OS, in PDAC.

Certainly, we should recognize that the present study has several limitations, such as the limited sample size and asymmetric distribution between cohorts. In fact, this limitation might be the underlying cause of not having sufficient statistical power to detect more significant associations within the data sets. This could be solved using a larger cohort where more accurate results could be obtained. It is worth noting that the sample size was mostly affected for the study of circulating immune cell populations, mainly due to the fact that these samples had to be processed fresh. In order to avoid potential biases, all these samples were collected at the same hospital. In contrast and taking into consideration the well-known biological redundancy between several analyzed cytokines, we developed a couple of scores trying to avoid the collinearity phenomenon. All of the above could be compensated by the increase of cohort’s follow-up period since the censored events could be reduced. In this sense, we certainly hope to validate externally our results in a new prospective and broader study. Finally, and although several molecular test approaches are highly recommended in the metastatic PDAC setting from the outset [47] there is a tremendous gap in these less advanced stages.

To summarize, we were able to find differentially expressed cytokines between different cohorts that may distinguish PDAC patients with different clinical stages. Furthermore, in the BL cohort two scores played an outstanding role as independent prognostic factors for PFS and OS. This study demonstrated that eotaxin-3, MDC, NT-3, MIP-3, and angiogenin may have a predictive role for *nab*-paclitaxel and gemcitabine treatment response, while oncostatin, BDNF, IP10, and NAP-2 for FOLFIRINOX. Finally, we demonstrated that circulating B and T lymphocytes monitoring, and their tumor infiltration, could be a useful tool to identify patients with worse outcome.

## 5. Conclusions

Late diagnosis, metastasis, and chemoresistance are the major challenges in the management of cancer patients. Here, we show that some lymphocyte populations and cytokines serve as prognostic and predictive biomarker candidates in patients diagnosed of resectable or borderline (BL) pancreatic adenocarcinoma (PDAC).

## Figures and Tables

**Figure 1 cancers-14-05993-f001:**
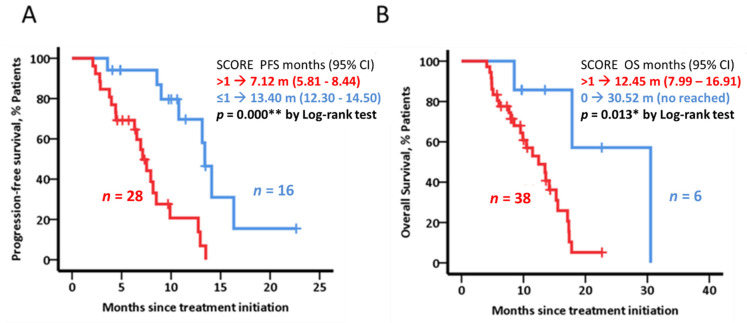
PFS and OS KM curves based on different cytokine score in BL patients (*n* = 44). (**A**) Kaplan–Meier plot represents PFS dichotomized depending on the points given by four cytokine levels, one point was given when cytokine levels were associated with worse PFS, 0 points (blue) between one and four (red). (**B**) The same as (**A**) for OS and dichotomized depending on the points given by four cytokine levels. In the text, PFS or OS and 95% CI values are shown. Log-rank test was used to test for statistical significance. * and ** in the figures indicate significant (*p* < 0.05) and very significant (*p* < 0.01) statistical differences.

**Figure 2 cancers-14-05993-f002:**
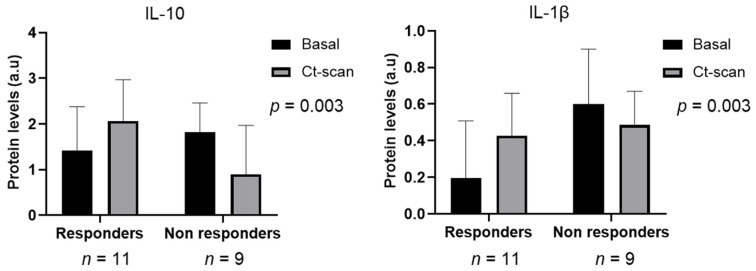
IL1-10 and IL-1β dynamics in responders and non-responders. Cytokine levels are represented in 2 different time-points (basal: black; after CT assessment; grey). General lineal model was used to test for statistical significance and both demonstrated very significant (*p* < 0.01) statistical differences.

**Figure 3 cancers-14-05993-f003:**
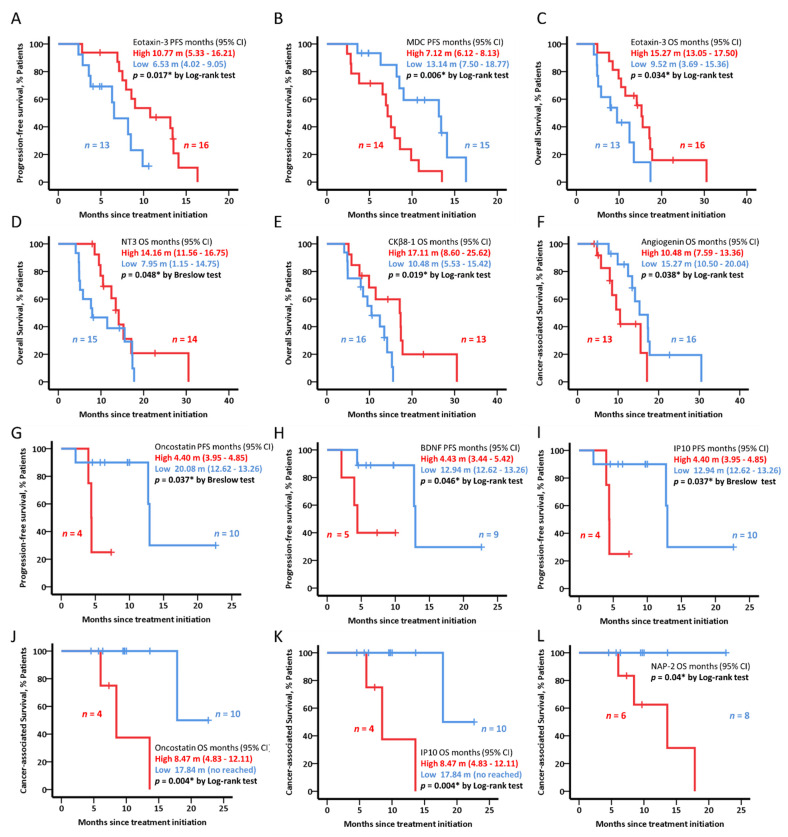
PFS/OS curves for baseline cytokine levels, BL patients (*n* = 29). Kaplan–Meier plots represent (**A**,**B**) PFS, (**C**–**E**) OS, and (**F**) cancer-associated death in BL patients receiving nab-paclitaxel and gemcitabine (*n* = 29) and (**G**–**I**) PFS, (**J**–**L**) cancer-associated death in BL patients receiving FOLFIRINOX stratified by the median baseline levels of each cytokine (*n* = 14). In the text, PFS or OS and 95% CI values are shown. Breslow or Log-rank test was used to test for statistical significance. * in the figures indicate significant (*p* < 0.05).

**Figure 4 cancers-14-05993-f004:**
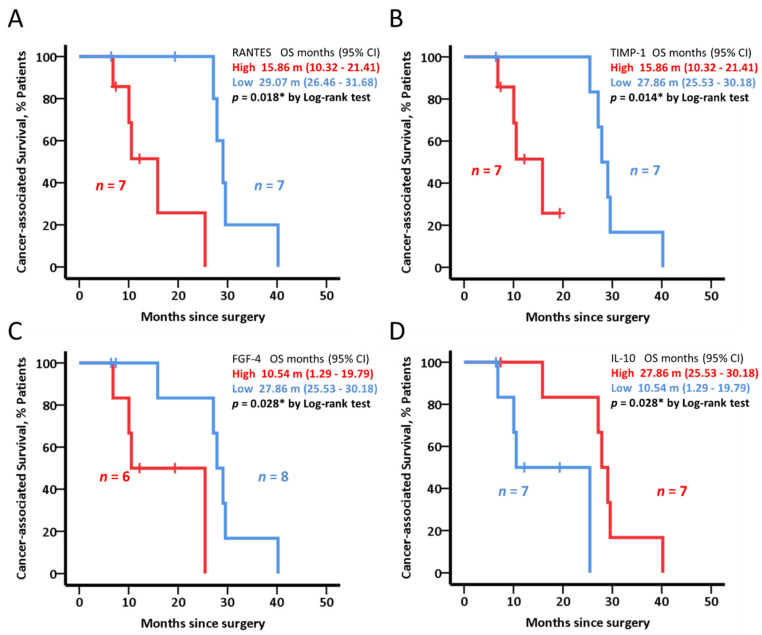
Cancer-associated survival analysis of baseline cytokine levels in resectable patients (n = 14). Kaplan–Meier plots represent cancer-associated death in resectable patients stratified by the median baseline levels of RANTES (**A**), TIMP-1 (**B**), FGF-4 (**C**) and IL-10 (**D**), lower than the median (blue) or higher than the median value (red). In the text, OS and 95% CI values are shown. Log-rank test was used to test for statistical significance. * in the figures indicate significant (*p* < 0.05), statistical differences.

**Figure 5 cancers-14-05993-f005:**
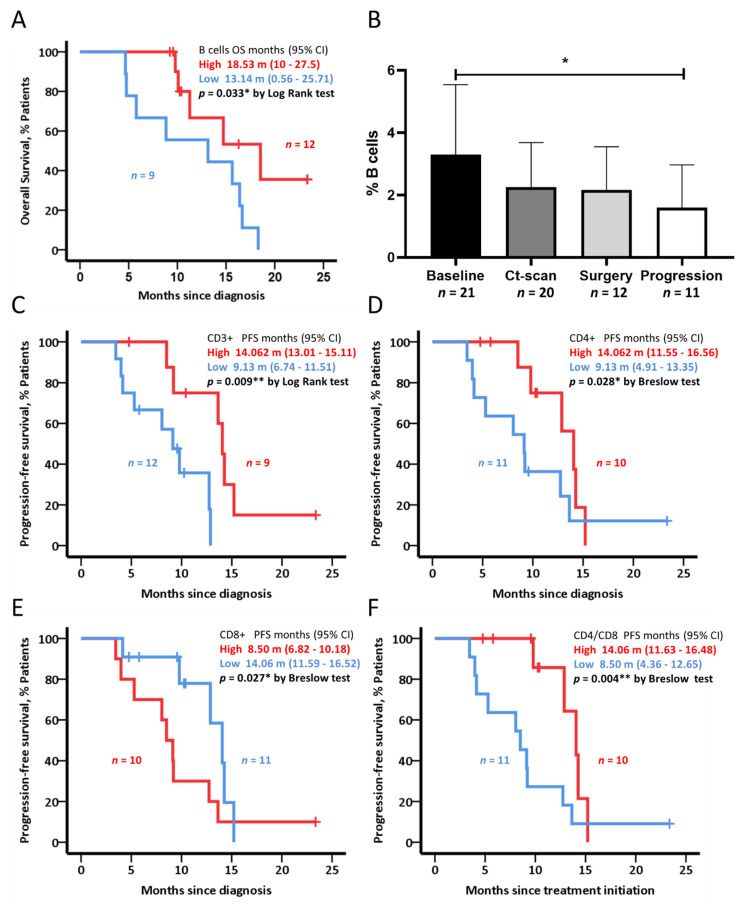
Circulating lymphocytes and clinical outcome in the BL cohort (*n* = 21). (**A**) Kaplan–Meier plots for B cells OS analysis. (**B**) B-cell population frequencies during patients’ follow-up in the BL cohort. (**C**–**F**) Kaplan–Meier plots for CD3, CD4, CD8, and CD4/CD8 PFS analysis. Data represented as mean ± s.d. * *p* < 0.05, *** p* < 0.01.

**Figure 6 cancers-14-05993-f006:**
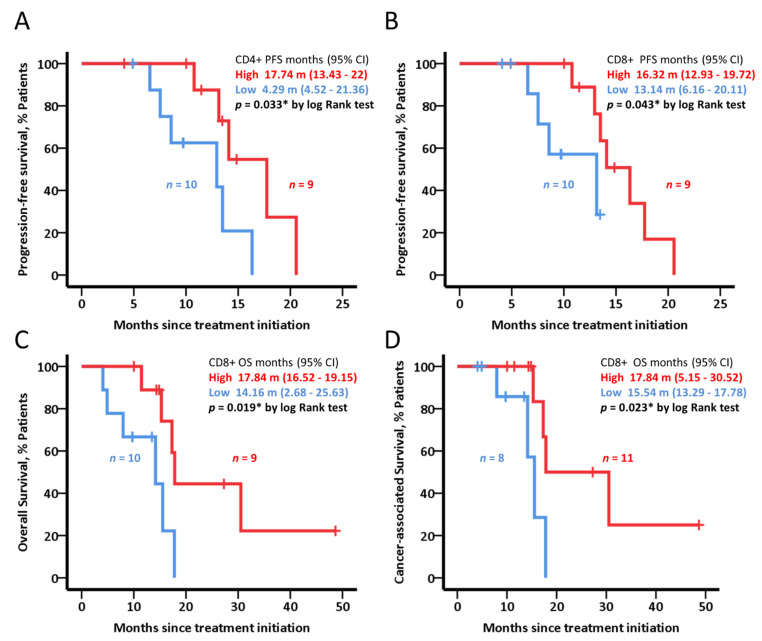
Correlation of CD4 and CD8 T-cell populations in tumor tissue, with clinical outcomes in the BL cohort (*n* = 19). (**A**,**B**) Kaplan–Meier plots for PFS, (**C**) for OS, and (**D**) for cancer-associated death. * in the figures indicate significant (*p* < 0.05).

**Table 1 cancers-14-05993-t001:** Clinical and demographic characteristics of BL and resectable patients.

Variable	BL Patients (*n* = 47) (%)	Resectable Patients (*n* = 17) (%)	Chi-Square Test
Median Age	66 (41–81)	65 (50–80)	0.84 ^T^
Gender			0.213 ^C^
Female	25 (53.2%)	12 (70.6%)	
Male	22 (46.8%)	5 (29.4%)	
Neoadjuvant therapy			
*Nab*-Paclitaxel-Gemcitabine	31 (66%)		
FOLFIRINOX	16 (34%)		
Neoadjuvant ChemoRT			
Yes	18 (38.3%)		
No	29 (61.7%)		
Neoadjuvant SBRT	7 (14.9%)		
Response			
Partial response	17 (36.2%)		
Stable disease	21 (44.7%)		
Progression	9 (19.1%)		
Surgery	29 (61.7%)	All	
Pathologic response			
0	1 (2.1%)		
1	5 (10.6%)		
2	10 (21.3%)		
3	5 (10.6%)		
Perineural invasion			0.203 ^F^
Yes	16 (34%)	15 (88.2%)	
No	6 (12.8%)	1 (5.9%)	
Vascular invasion			0.584 ^C^
Yes	11 (23.4%)	10 (58.8%)	
No	11 (23.4%)	7 (41.2%)	
R			0.152 ^F^
R0	16 (34%)	15 (88.2%)	
R1	8 (17%)	2 (11.8%)	
Lymph nodes involved			0.505 ^F^
Yes	15 (31%)	13 (76.5%)	
No	8 (17%)	4 (23.5%)	
pT			0.005 **^F^
0	1 (2.1%)	0 (0%)	
1	9 (19.1%)	2 (11.8%)	
2	10 (21.3%)	5 (29.4%)	
3	3 (6.4%)	9 (52.9%)	
4	0 (0%)	1 (5.9%)	
pN			0.677 ^F^
0	9 (19.1%)	4 (23.5%)	
1	9 (19.1%)	1 (64.7%)	
2	5 (10.6%)	2 (11.8%)	
Adjuvant treatment			0.052 ^F^
Yes	14 (29.8%)	16 (94.1%)	
No	8 (17%)	1 (5.9%)	
Adj chemoRT			1 ^F^
Yes	1 (2.1%)	1 (5.9%)	
No	18 (38.3%)	16 (94.1%)	

Abbreviations: ChemoRT: concomitant chemotherapy and radiotherapy; SBRT: stereotactic body radiation therapy; R: residual tumor; R0: no cancer cells seen microscopically at the primary tumour site; R1: cancer cells present microscopically at the primary tumour site; pT: pathological tumor size AJCC 8th edition, being ypT for BL and pT for resectable patients; pN: pathological lymph node staging AJCC 8th edition. Statistical analyses were two-tailed unpaired Student t test, chi-squared test and bilateral Fisher exact test when there were less than five individuals in a group, they are indicated as superscripts T, C, or F respectively on *p* values. ** indicate very significant (*p* < 0.01) statistical differences.

## Data Availability

The datasets generated and/or analyzed during the current study are not publicly available due to risk of personal information leakage but are available from the corresponding author on rea-sonable request.

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
