# Peer review of "Cytokines and Lymphoid Populations as Potential Biomarkers in Locally and Borderline Pancreatic Adenocarcinoma"

_cancers, 2022, doi:10.3390/cancers14235993_

Round 1
Reviewer 1 Report (Previous Reviewer 2)
Thank you for the revisions. I am sure the article will be of interest to the readers. No objections.
Author Response
Thank you for taking your time to review our article. We really appreciate your inputs.
Reviewer 2 Report (Previous Reviewer 1)
This manuscript mostly describes cytokine levels/changes between different groups of PDAC patients and proposes some to use as biomarkers of response and progression. There are still some minor grammatical errors, but the English writing has been improved substantially in this revision. A few sections could still use clarification.
Materials and methods
1. More detail on how the cytokine arrays are quantified would be useful, since they make up such a large portion of the manuscript. Were all arrays done at the same time, and if not, how were the results normalized?
2. In section 2.3, line 155, what is the reference 190?
3. How were IHC stains analyzed/quantified?
Results
4. In Figure 2, when was the radiological assessment done? Did all of the patients analyzed here get surgery beforehand? In the comparisons showing IL10 and IL1B were underexpressed in R patients, are you only considering the basal timepoint?
5. How is a low vs high cytokine level determined for Figures 3 and 4?
6. In section 3.5, how did you determine a high vs low level of T cell subsets? Please include definitions of subsets here as well (such as for CD4+ TID).
7. In Figure 5, is % B cells a percentage of all viable cells? Please add this information to other plots showing immune cell percentages as well including Supplemental Figure 3.
8. Where is Supplementary Figure 2 referenced in the text of the Results section?
Discussion
9. In line 349, which figure shows BL patients with lower IL10 have longer PFS? I don't see it in Figure 3 with the other BL data.
Author Response
- More detail on how the cytokine arrays are quantified would be useful, since they make up such a large portion of the manuscript. Were all arrays done at the same time, and if not, how were the results normalized?
As suggested by the reviewer, a shortened protocol of the cytokine array quantification and analysis has been included in section “2.2. Human cytokine antibody array” from Material and Methods (pages 5-6).
Cytokine arrays from serum were performed using samples containing a final protein concentration of 100ug. The appropriate concentration was previously determined by incubating several membranes with different amounts of protein concentration. 100 ug was the best condition for detecting all cytokines/ growth factors in the array without reaching saturation.
All arrays were not performed at the same time since it was technically not possible to handle all membranes at the same time. Thus, arrays were performed in groups of 18 each time. Each array contains negative and positive controls. Negative controls must show no signal after membrane development, and the positive controls, serve to normalize all the membranes, compared to the mean signal density of positive controls in the healthy donors group.
The protocol was performed according to the manufacturer’s recommendations. Briefly, the protein content of serum samples was quantified using Bradford reagent. Membranes were blocked with blocking buffer at room temperature (RT) for 30 min and incubated overnight at 4 °C with 100 µg of serum protein diluted in 1ml blocking buffer. The next steps include two overnight incubations, at 4°C, with biotinylated anti-cytokine antibody mix following HRP-conjugated streptavidin, with serial washing of the membranes between each incubation step. Finally, membranes were incubated with detection buffer for 2 min and revealed by ChemidocTM MP Imaging System (Bio-Rad). Signals were quantified using ImageLab software (Bio-Rad) and normalized using the six positive controls present on each membrane, using the mean of a control group to normalize the array data. To normalize the values, the following calculus was performed:
X(Ny)= X(y) * P1 /P(y)
Where: X (Ny) = normalized signal intensity for spot “x” on array “y”; X(y) = mean signal density for spot “x” on array for sample “y”; P1 = mean signal density of 6 positive control spots of healthy donors; P(y) = mean signal density of positive control spots on array “y”. After this normalization, the relative expression levels of each cytokine in our samples of interest were compared. For statistical analysis Perseus software (Version 1.6.5) was used, where the previously obtained normalized data “log2” was applied. Then the data were normalized with adjustment, and the correspondent statistical test as two-way Student´s t-test between groups was performed. The p-values and fold changes obtained were then represented with the Scatter Plot tool.
- In section 2.3, line 155, what is the reference 190?
We apologize for the error. The reference 190 has been deleted from line 155.
- How were IHC stains analyzed/quantified?
The IHC stains were analyzed by expert pathologists, following the criteria used in the clinical setting. Briefly, all IHC were observed under the microscopy and three intensity groups were considered, according to the area of CD4 or CD8 infiltration. The categories were the following: low, intermediate and high infiltration. According to your comment, we have added this part to the manuscript.
Results
- In Figure 2, when was the radiological assessment done?
The radiological assessment was performed after neoadjuvant treatment, after 2-3 months of this neoadjuvant treatment. Based on your comment, we have clarified it in the text.
Did all of the patients analyzed here get surgery beforehand?
The basal measurement refers to the cytokine levels prior receiving the neoadjuvant treatment, and the Ct-scan measurement refers to the cytokine levels after the neoadjuvant treatment, when the radiological assessment was done. None of the patients had undergone surgery when these samples were collected.
In the comparisons showing IL10 and IL1B were underexpressed in R patients, are you only considering the basal timepoint?
Indeed, this analysis was only considering the basal timepoint.
- How is a low vs high cytokine level determined for Figures 3 and 4?
First, the median value for each cytokine and each timepoint was determined and considered as the cut-off value. Levels below the cut-off value were considered as “Low”, and levels above the cut-off value as “High”,as was just explained in the Material and Methods Section.
- In section 3.5, how did you determine a high vs low level of T cell subsets? Please include definitions of subsets here as well (such as for CD4+ TID).
High and low levels of T cell subsets were determined in the same way as cytokines. After calculating the median value of each T cell subset in each timepoint (cut-off value), “low” and “high” were assigned to levels below and above this cut-off value, respectively, as was previously explained in the Material and Methods Section.
T cells were classified according to CD27/CD28 expression markers into poorly-differentiated (CD27+ CD28+, TPD), intermediately-differentiated (CD27negative CD28+, TID) and highly-differentiated (CD27negative CD28low/negative, THD) subsets, as is explained in Part 2.3 of Material and Methods Section.
These cell subsets‘ definitions were based on the reference (https://onlinelibrary.wiley.com/doi/10.1002/cyto.a.22351), where T cell markers' were summarized according to their functionality. CD27 and CD28 markers were selected in order to simplify the flow cytometry analysis.
- In Figure 5, is % B cells a percentage of all viable cells? Please add this information to other plots showing immune cell percentages as well including Supplemental Figure 3.
B cells refer to B lymphocytes, defined as CD11b- CD19+, as depicted in “2.3. Flow cytometry” from Material and Methods (page 6). The analysis of B lymphocytes in healthy donors and borderline patients was not significant, and thus we did not include the data in Supplemental Figure 3. We show here the data.
- Where is Supplementary Figure 2 referenced in the text of the Results section?
We apologize for the mistake, we consider that this figure is not necessary and thus we will not include it in the final version of the article.
Discussion
- In line 349, which figure shows BL patients with lower IL10 have longer PFS? I don't see it in Figure 3 with the other BL data.
The figure showing that BL patients with lower IL10 have longer PFS is not shown in the manuscript, as it is explained in the text (page 9). Instead a score including IL-10, MDC, MIF and Eotaxin-3 for PFS analysis is shown in Figure 1A (page 10).
Here we show the individual graph for each cytokine in the score from Figure 1A.

This manuscript is a resubmission of an earlier submission. The following is a list of the peer review reports and author responses from that submission.
Round 1
Reviewer 1 Report
The authors present a study on assessing potential biomarkers in cytokines and circulating/tumor-infiltrating immune cells that will help predict pancreatic cancer management and prognosis. Unfortunately, the manuscript is very difficult to read, since the references to figures in the text did not work. All references instead say " Error! Reference source not found." I can't review the manuscript properly until this is fixed. In general, the writing would also benefit from extensive editing for clarity/wordiness and English grammar.
Author Response
We really appreciate and are grateful for the effort and quality of reviewer 1's considerations. Thanks to these considerations, we have tried to significantly improve the quality of the English of the article thanks to its review by an English speaker co-author. We believe that this makes the article easier to read as reviewer 1 requested. Hopefully this time the references to the figures will work, in any case, we do not consider this to be a problem for which we are entirely responsible.
Reviewer 2 Report
González-Borja et al are presenting the findings of their clinical trial to identify biomarkers for lower stage pancreatic cancer. To achieve this the authors studied 64 patients total of which 47 were defined borderline and 17 resectable. The study has several weaknesses. It remains unclear what the statistical power is. Some of the figure presentations are unclear: e.g. Fig 2 both figures have a p value of 0.03 for what responders vs non responders,basal versus Ct scan? This is unclear. Have you utilized a biostatistican, if not, please do so as your data will benefit from it. Your patient cohort is very variable,age, gender etc and in addition no words about significant accompanying disease like renal impairment etc. The patient numbers included were not stated in the methods section at all only in the table. It remains unclear who defined "borderline" and what is borderline in international guidelines? You are describing IHC and flow cytometry yet there are no images in the paper but in the supplementary. Why? This should be changed. In addition, those images have no scale bars. And the figure legend is very very poor.
There are several language issues may it be due to language barrier or just being a bit sloppy (I am sorry) where multiple times prepositions are missing - e.g. introduction third line a "to" is missing.
Author Response
We can only be grateful for the effort of your review that will hopefully result in a significant improvement of the current version. We have modified the method section including a particular section about statistical power and how sample size was calculated with the help of our biostatistician. Similarly, we have tried to fix those issues you pointed out about figure 2. We consider our cohort is totally representative of subjects with pancreatic adenocarcinoma. We have adopted, as in most similar articles, the representation of patients recruited according to their ECOG, trying to ensure that those patients with EGOG equal or higher than 2 did not receive the treatments used here, as recommended by the ESMO, ASCO and NCNN guidelines. As we referred previously, reference of how sample size was calculated has been added in the method section reason why we consider more properly to keep the final number of subjects enrolled in the result section.
Currently, there is not still an universal consensus about BL definition, being at least 4 so far (NCCN, Alliance, MD Anderson and AHPBA/SSO/SSAT). We enrolled subject considered as BL based on NCCN criteria with poor prognostic factors (suspected of micrometastatic disease, risk of positive margin during surgery and/or CA 19.9 ≥ 300 UI/ml without jaundice). Based on your recommendation, both have been added to the current version. Finally, we have followed your recommendations including pictures about IHC and flow cytometry approaches and trying to improve figure legends in the same way that we have tried to significantly improve the quality of the English of the article thanks to the its review by an English speaker co-author.